# Predictors of CrossFit Open Performance

**DOI:** 10.3390/sports8070102

**Published:** 2020-07-20

**Authors:** Gerald T. Mangine, Joy E. Tankersley, Jacob M. McDougle, Nathanael Velazquez, Michael D. Roberts, Tiffany A. Esmat, Trisha A. VanDusseldorp, Yuri Feito

**Affiliations:** 1Department of Exercise Science and Sport Management, Kennesaw State University, Kennesaw, GA 30144, USA; jlipham1@students.kennesaw.edu (J.E.T.); jmcdou11@students.kennesaw.edu (J.M.M.); nvelazqu@students.kennesaw.edu (N.V.); tesmat@kennesaw.edu (T.A.E.); tvanduss@kennesaw.edu (T.A.V.); yfeito@kennesaw.edu (Y.F.); 2School of Kinesiology, Auburn University, Auburn, AL 36849, USA; mdr0024@auburn.edu

**Keywords:** high intensity functional training, athlete, critical power, aerobic capacity, ultrasound

## Abstract

The 2018 CrossFit Open (CFO) was the initial stage of an annual competition that consisted of five weekly workouts. Current evidence suggests that a variety of fitness parameters are important for progressing beyond this stage, but little is known about which are the most important. To examine relationships between CFO performance, experience, and physiological fitness, sixteen experienced (>2 years) athletes (30.7 ± 6.9 years, 171 ± 12 cm, 78.0 ± 16.2 kg) volunteered to provide information about their training and competitive history, and then complete a battery of physiological assessments prior to competing in the 2018 CFO. Athletes’ resting energy expenditure, hormone concentrations, body composition, muscle morphology, cardiorespiratory fitness, and isometric strength were assessed on two separate occasions. Spearman correlations demonstrated significant (*p* < 0.05) relationships between most variables and performance on each workout. Stepwise regression revealed competition experience (R^2^ = 0.31–0.63), body composition (R^2^ = 0.55–0.80), vastus lateralis cross-sectional area (R^2^ = 0.29–0.89), respiratory compensation threshold (R^2^ = 0.54–0.75), and rate of force development (R^2^ = 0.30–0.76) to be the most common predictors. Of these, body composition was the most important. These fitness parameters are known targets with established training recommendations. Though preliminary, athletes may use these data to effectively train for CFO competition.

## 1. Introduction

Several CrossFit^®^ competitions occur throughout the year at the local, regional, national, and international levels. Similar to the training strategy [1,2], each competition involves a series of workouts that variably require some combination of strength, cardiorespiratory, and gymnastic skill. The ultimate, annual competition is known as the Reebok CrossFit Games™ (the Games) and its winners are awarded the title of “Fittest on Earth™”. Although several avenues exist for athletes to earn a spot in the Games, as of 2018 the CrossFit^®^ Open (CFO), an international online qualifier, was the most popular [3,4]. The CFO is a 5-week competition where athletes are given four days to complete each week’s workout(s) at their normal training facility and submit their best score, which was either certified by an official judge or recorded as a video. Scores are commonly entered as repetitions completed, a completion (of prescribed work) time, weight lifted, or some combination of these. Regardless, athletes’ scores are then ranked on each week using a system where the best performance receives the lowest rank (i.e., 1st place). Athletes’ ranks are tallied over the course of the CFO, and those who rank high enough after five weeks (i.e., have the lowest sum of their ranks from each workout) will advance in the competition. Since hundreds of thousands of athletes participate in the CFO each year [3], advancement is very difficult; less than 1% will qualify. The high amount of variability in workout requirements further adds to the difficulty because athletes must possess sufficient skill across several fitness parameters. However, athletes who train to simultaneously maximize performance in all possible parameters may create a “blender effect” [5] and inadvertently lower their chances of success. Instead, it may behoove these athletes to sequence training to build towards the most important training factors or group training targets based on their compatibility. Currently, however, limited data exist to assist athletes in identifying the most important parameters to focus on during training [6,7,8,9,10,11,12,13,14].

Half of studies relating physiological fitness to CrossFit^®^ performance have been completed solely within a laboratory setting [6,8,9,11]. Butcher and colleagues (2015) reported significant relationships between performance during common benchmark workouts and several physiological measures collected during anthropometric testing, a graded exercise test, a Wingate Anaerobic Threshold test, and the CrossFit^®^ total (i.e., sum of 1-repetition maximums in the deadlift, back squat, and overhead press) in adults with approximately 3.9 years of CrossFit^®^ experience. However, only CrossFit^®^ total strength could significantly predict performance in two of the three benchmark workouts. Martinez-Gomez et al. (2019) also found strength and power (in the back squat) to be related to performance in a group of experienced (≥one year) CrossFit^®^ practitioners. In this case, their performance in the workouts of the 2017 CFO completed over five consecutive days. In contrast, Bellar et al. (2015) found that aerobic capacity and Wingate performance were predictive of performance during a novel, 12-min workout using an ‘as many repetitions as possible’ (AMRAP) scheme, but not in another novel workout using a set repetition scheme (15-12-9 repetitions) that had to be completed as fast as possible. Further, they found that the best predictor for both workouts was CrossFit^®^ experience (i.e., possessing at least one year of experience or having no experience). Later, Feito and colleagues (2018) found that the ability to maintain work and recover between four consecutive Wingate trials separated by 90 s rest was the best predictor of another novel workout (15-min AMRAP) in adults with CrossFit^®^ experience. Experience in this study was defined as having participated in CrossFit^®^ training for at least two years and the ability to complete the benchmark workout “Fran” within 3 (men) or 4 (women) minutes. Collectively, these studies suggest that a variety of physiological measures may influence performance, depending on the workout and experience. However, it is difficult to generalize these findings to competition, because workouts were performed in a controlled laboratory setting and each study differentially accounted for experience. Although athletes in these studies have ranged in competitive status, from recreational to advanced, laboratory workouts do not adequately emulate the competitive setting and may affect the physiological characteristics that are relevant to competition [15,16,17,18]. Therefore, to better assist in the preparation efforts of coaches and athletes, it appears to be important to base the predictive ability of various physiological measures on their performance within the competitive setting. Moreover, a more comprehensive approach to accounting for experience (i.e., considering both years of experience and performance) seems to be most prudent.

Less than a handful of studies exist that help provide information on the physiological aspects that are important to competition performance [7,10,13,14]. Recently, Martínez-Gómez and colleagues (2020) directly measured a variety of fitness parameters, and concluded that their predictive ability was primarily dependent on the specific workout but overall, best explained by the combination of jumping power and aerobic capacity. Although this conclusion provides some clarity for developing training recommendations, their generalizability is negatively impacted without a clear description or evidence of the participants skill as CrossFit^®^ competitors (i.e., actual performance in each workout and past competitive success were not reported). In contrast, studies that have provided clear evidence of competitive success are limited by the validity of their data [7,10,13]. Serafini and colleagues (2018) observed that higher ranking competitors of the 2016 CFO outperformed their lower-ranking counterparts (all within the top 1500) in most fitness indices, though measures of strength (i.e., back squat, deadlift) and power (i.e., clean and jerk, snatch) provided the clearest distinctions. A similar examination was performed in regional (no longer a stage in the competition) athletes of the same year [10], but relationships were only found in women and select fitness measures (i.e., clean and jerk, snatch, “Filthy-50”, and 400-m sprint). One year later, Schlegel et al. (2020) interviewed the Top 20 Czech athletes from 2019 and found relationships between ranking and nearly every measure, except for the athletes’ reported scores in the back squat, deadlift, and pull-ups. The primary limitation with each of these studies was that, aside from competitive ranking, these data were self-reported. The accuracy and timeliness (i.e., the proximity of reported scores with competition performance) of data collected in this manner is difficult to determine and, thus, make training recommendations with sufficient confidence. Nevertheless, because the dependent variables within these studies was defined by competitive success, these data provide the most direct path towards identifying measures that are influential of competitive achievement. Therefore, the purpose of this investigation was to determine the influence of experience, as well as self-reported and measured physiological fitness variables, on CFO performance. Based on previous evidence [6,7,8,9,10,11,12,13,14] and the general aim of CrossFit^®^ training to promote general physical preparedness [1,2], we hypothesized that relationships would exist between CFO performance, experience, and physiological fitness variables across each of the investigated parameters (i.e., body composition, resting hormone concentrations, resting energy expenditure, cardiorespiratory fitness, and muscular strength). However, we also hypothesized that the influence of each physiological fitness variable on performance would vary between workouts and be modified by experience.

## 2. Materials and Methods

### 2.1. Participants

Sixteen experienced (>2 years) male (n = 8) and female (n = 8) CrossFit^®^ practitioners (30.7 ± 6.9 years, 171 ± 12 cm, 78.0 ± 16.2 kg), who had volunteered to participate in a larger investigation into the physiological differences among CrossFit^®^ athletes and physically-active adults [19], were drawn for the present study. All athletes were free of any physical limitations (determined by medical and physical-activity history questionnaire and PAR-Q+), had been regularly (>three sessions per week) participating in resistance and CrossFit^®^ training for a minimum of two years, and had signed up to compete in the 2018 CFO upon their own initiative. All athletes had previous CFO experience (1–6 years) and, during that time, the highest rank achieved by the athletes ranged between 19th–46,013th place, within their respective competitive division. Six athletes possessed 1–4 years of experience at the regional round of the competition, with an additional two athletes reaching that round in 2018. Five of the athletes also possessed 2–4 years of experience at the Games, and three additional athletes reached this round in 2018. The remaining athletes (n = 8) had never progressed beyond the CFO. Following an explanation of all procedures, risks and benefits, each athlete provided his or her written informed consent to participate in the study. The study was conducted in accordance with the Declaration of Helsinki, and all methods had been previously approved by the University’s Institutional Review Board (#16-215 and #17-501).

### 2.2. Study Design

For this cross-sectional study, athletes who were enrolled to compete in the 2018 CFO provided details regarding their training and competitive history and then completed physiological assessments across two visits to the Exercise Physiology Laboratory. Both visits occurred within one month of the 2018 CFO and were separated by 3–7 days. On the first visit, each athlete provided a resting blood sample, before completing ultrasound assessments of muscle morphology and a graded exercise test. The athletes began the second visit with a resting metabolic energy expenditure, followed by measures of body composition, isometric mid-thigh pull strength, and a 3-min maximal cycling sprint. All testing sessions occurred in the morning (~6:00–10:00 a.m.) with the athletes having abstained from unaccustomed physical activity and alcohol for 24 h, caffeine for 12 h, and fasted for 8 h. Athletes completed all measurements while wearing comfortable athletic clothing, and were able to consume a light snack prior to performance testing. Prior to leaving the laboratory on the first visit, the athletes were asked to complete a 24-h dietary recall, retain a copy, and follow a similar diet prior to their second visit. Following both laboratory visits, each athlete completed 2018 CFO workouts on their own time (within competition parameters) at their normal training facility, and uploaded their best score to the competition website [20]. Relationships were then examined between 2018 CFO performance and experience, as well as all self-reported and measured fitness outcomes.

### 2.3. Competition Performance and Self-Reported Fitness

Athletes who participate in the CFO must create a publicly-available online profile [20] to upload performance scores on each week of the competition. There, competition representatives verify the accuracy of submitted scores, before updating their status as being official. In addition to this function, each athlete’s profile maintains a history of their final ranking at each stage of the Games competition for each year they were enrolled as a competitor. The athlete may also upload their personal records for a variety of fitness measures. For the present study, 2018 CFO workout (18.1–18.5) performance and description data (Table 1), years of experience (i.e., number of years enrolled as a competitor) in the CFO, in the regional round (individual, team, and total), and in the Games (individual, team, and total), 2017 CFO ranking, highest ever CFO ranking, and personal records in the back squat, deadlift, clean, and jerk, snatch, “Fran”, “Grace”, and “Helen” were collected from each athlete’s profile. Although there were occasional instances where an athlete had recorded their score for other available fitness parameters, most athletes in the present sample left these blank on their profile.

### 2.4. Resting Physiological Measures

#### 2.4.1. Blood Sampling and Biochemical Analysis

Resting hormone concentrations are known to be affected by training, and may indirectly provide evidence of an athlete’s recovery status [18], and, thus, may act as a modifier on their performance in competition. To quantify resting concentrations in relevant hormones, blood samples were collected from each athlete on the morning of their first visit and before any physical activity. Commercially available, enzyme-linked immunosorbent assays were used to quantify circulating concentrations of testosterone (ng·dL^−1^), cortisol (μg·dL^−1^), and insulin-like growth factor 1 (IGF-1; ng·mL^−1^) within the samples using a 96-well spectrophotometer (BioTek, Winooski, VT, USA). Additionally, the ratio of testosterone to cortisol concentrations (TC ratio × 1000) was calculated as a variable of interest. The collection and biochemical analysis of blood samples have been described in greater detail elsewhere [19].

#### 2.4.2. Resting Energy Expenditure

Resting energy expenditure may be affected by an athlete’s body composition, as well as their dietary, nutritional, and training habits [21], all of which may have a modifying effect on athletic performance. Upon their arrival on visit two, and after having adhered to recommended pre-testing procedures [22], the athletes were asked to lay supine for 30 min within a minimally lit (i.e., only light from the computer), quiet room, and under a ventilated hood. The hood was connected to a metabolic measurement system (Parvo Medics TrueOne 2400, ParvoMedics Inc., Salt Lake City, UT, USA) to quantify resting energy expenditure (kcals·day^−1^) within a 5-min interval of measured volume of oxygen consumption (VO_2_), which produced a coefficient of variation of less than 10% [22]. The collection of this measure has been described in greater detail elsewhere [19].

### 2.5. Body Composition

#### 2.5.1. Muscle Morphology

Ultrasound images were taken from the right rectus femoris, vastus medialis, vastus lateralis, biceps brachii, and triceps brachii, using a 12 MHz linear probe scanning head (General Electric LOGIQ S7 Expert, Wauwatosa, WI, USA), coated with water soluble transmission gel. Two consecutive images were collected with the probe oriented longitudinally in Brightness Mode (B-mode), while another two consecutive images were collected using a panoramic sweep in the extended field of view mode [23]. All images were transferred to a personal computer for analysis via Image J (National Institutes of Health, Bethesda, MD, USA, version 1.45s). Muscle thickness (±0.01 cm) was quantified from all B-mode images. Muscle cross-sectional area (CSA; ±0.1 cm^2^) and echo intensity in arbitrary units (au) were quantified from all panoramic images [23,24], with mean echo intensity values being corrected for subcutaneous fat thickness (SFT; averaged from the SFT values obtained at the medial, midline, and lateral sites of each muscle) using Equation (1) [25]. The same investigator performed all landmark measurements, positioned each participant, and collected and analyzed all images. Ultrasound image collection procedures have been described in greater detail elsewhere [19].
Corrected echo intensity (EI) = Raw echo intensity + (SFT × 40.5278)(1)

#### 2.5.2. Body Fat Percentage

Athlete height (±0.1 cm) and body mass (±0.1 kg) were initially determined using a stadiometer (WB-3000, TANITA Corporation, Tokyo, Japan) and used to calculate body mass index (BMI; ±0.1 kg·m^−2^). Body composition was estimated using a 4-comparement model [19,26] and data extracted from dual energy X-ray absorptiometry (iDXA, Lunar Corporation, Madison, WI, USA), air displacement plethysmography (BodPod, COSMED USA Inc., Chicago, IL, USA), and bioelectrical impedance analysis (770 Body Composition and Body Water Analyzer, InBody, Seoul, South Korea). Subsequently, body fat percentage (BF%), fat mass (±0.1 kg), and fat-free mass (±0.1 kg), body density (kg·L^−1^) from the BodPod, and regional (arms, legs, and trunk) estimates of bone mineral content (±0.1 kg) and non-bone lean mass (±0.1 kg) obtained from iDXA were used as variables of interest. The details of body composition assessment have been described elsewhere [19].

### 2.6. Cardiorespiratory Fitness Assessments

#### 2.6.1. Graded Exercise Test

Athletes were asked to wear a heart rate (HR) monitor (Team^2^, Polar, Lake Success, NY, USA), a nose clip, and a 2-way valve mask connected to a metabolic measurement system (True One 2400, ParvoMedics Inc., Salt Lake City, UT, USA), while they completed a continuous, ramp exercise protocol on an electromagnetic-braked cycle ergometer (Lode Excalibur Sport, Lode., B.V., Groningen, The Netherlands). The protocol required the athletes to complete a 3-min warm-up against a 50 W resistance. Immediately following the warm-up period, the resistance was increased to 75 W, and the athletes were asked to maintain a self-selected pedaling rate that was greater than 50 RPMs. The assessment continued by increasing power output by 25 W every minute, and until volitional fatigue or pedaling rate dropped below 50 rpm’s for longer than 15 s. The procedures used to estimate peak aerobic capacity (VO_2peak_; mL·kg^−1^·min^−1^), respiratory compensation threshold (RCT; mL·kg^−1^·min^−1^), and gas exchange threshold (GET; mL·kg^−1^·min^−1^) have been previously described elsewhere [19]. Additionally, HR was recorded on each minute of a 50 W recovery period where the athletes cycled at their own cadence for three minutes. HR collected after 1 min (HR_Recovery_) was retained as an absolute value (bpm), to be used as a variable of interest. Relative values for HR_Recovery_, as a percentage of HR_Peak_, as well as RCT and GET, as percentages of VO_2Peak_, were also used as variables of interest.

#### 2.6.2. 3-min Maximal Cycling Sprint

Peak power (±1 W), critical power (CP; ±1 W) [27], and anaerobic work capacity (AWC; ±0.1 kJ) [28] were calculated from the athletes’ performance during a 3-min, maximal sprint on an electromagnetic-braked cycle ergometer (Lode Excalibur Sport, Lode., B.V., Groningen, The Netherlands) and retained as variables of interest. Briefly, this assessment occurred immediately following a standard warm-up and a 1-min baseline period where the athletes cycled for 55 s at 90 RPMs against no resistance. The athletes then accelerated to approximately 110 RPMs for 5 s, before entering the 3-min testing period. Throughout the entire test, athletes were instructed to maintain their highest possible cadence against a linear mode resistance, set at a power output equal to halfway between the VO_2_peak and GET, divided by the preferred cadence of untrained cyclists (70 RPM^2^) [27,28,29]. The procedures for this assessment have been described in greater detail elsewhere [19,30].

### 2.7. Isometric Mid-Thigh Pull Strength

Athletes completed the isometric mid-thigh pull test within a power rack (Rogue Fitness, Columbus, OH, USA) while standing upon a portable force plate (Accupower, AMTI, Watertown, MA, USA). Briefly, athletes were instructed to step up to an immobilized barbell and assume their preferred second pull, power clean position. Upon the researcher’s “3, 2, 1, Go!” command, the athletes were instructed to pull upwards on the barbell as hard and as fast as possible for a period of 6 s. All athletes were allotted three maximal attempts separated by approximately 3 min of rest. Ground reaction forces measured by the portable force plate were used to calculate peak force (F; in N), peak and average rate of force development (RFD_Peak_, RFD_AVG_; in N·s^−1^), and RFD across specific time bands (i.e., 0–30, 0–50, 0–90, 0–100, 0–150, 0–200, and 0–250 ms), using established methods [31]. Procedures related to the warm-up, bar positioning, and grip allowed for this assessment have been described in greater detail elsewhere [19].

### 2.8. Statistical Analysis

The results of the Shapiro-Wilk tests indicated that the assumption of normality was not met for several variables. Therefore, relationships between all independent variables and 2018 CFO performance were quantified by calculating Spearman’s Rho (ρ) correlation coefficients. The strength of observed relationships were interpreted using the following criteria: trivial (<0.10), small (0.10–0.29), moderate (0.30–0.49), high (0.50–0.69), very high (0.70–0.90), or practically perfect (>0.90) [32]. To determine the best predictors of performance, variables were initially grouped into categories (i.e., training and competitive experience, self-reported fitness, body composition, muscle morphology, resting hormone and energy expenditure, cardiorespiratory testing measures, and isometric mid-thigh pull strength). Then, the best predictor from each category for each workout was determined via stepwise regression. Subsequently, stepwise regression was repeated using only the best predictors identified from each category to determine the best overall predictor for each workout. A criterion alpha was set at *p* ≤ 0.05. All statistical analyses were performed using SPSS (v. 26.0, SPSS Inc., Chicago, IL, USA). All data are reported as mean ± standard deviation.

## 3. Results

Upon conclusion of the 2018 CFO, the athletes of the present study finished between 177th and 149,797th place (21,901 ± 37,035th) within their respective competitive divisions. Their performances and worldwide ranks for each workout are presented in Table 1.

### 3.1. Self-Reported Fitness and Competition Performance

#### 3.1.1. Training and Competition Experience

Several methods of quantifying experience were significantly (*p* < 0.05) related to 2018 CFO performance (Table 2). Of these, the highest rank ever achieved during any previous CFO was the best predictor of 18.1 (R^2^ = 0.59, *p* = 0.001). The number of years of CFO experience was the best predictor for 18.2a (R^2^ = 0.31, *p* = 0.024) and 18.3 (R^2^ = 0.45, *p* = 0.004), while the total amount of years of past experience in regional competition best predicted 18.4 (R^2^ = 0.52, *p* = 0.002) and 18.5 (R^2^ = 0.36, *p* = 0.013). The number of years of regular resistance training experience best predicted 18.2b (R^2^ = 0.63, *p* < 0.001).

#### 3.1.2. Self-Reported Fitness

Helen time was negatively (*p* < 0.05) associated with 18.1 and 18.4 performance, while all measures of strength were positively (*p* < 0.05) related to 18.2b performance. No other relationships were observed (Table 2). According to stepwise regression analysis, reported maximal deadlift was the best predictor of 18.2b performance (R^2^ = 0.98, *p* = 0.002) and Helen time explained 91% of variance in 18.4 performance (*p* = 0.012).

### 3.2. Resting Hormone Concentrations and Energy Expenditure

Aside from TC ratio (30.7 ± 31.1 [3.2–101.2]) being positively associated with 18.2b performance (ρ = 0.60, *p* = 0.024), resting hormone concentrations were not related to 2018 CFO performance. Likewise, resting energy expenditure (1777 ± 325 kcal·day^−1^; range = 1252–2376 kcal·day^−1^) was not related to performance on any workout.

### 3.3. Body Composition

#### 3.3.1. Body Fat Percentage Assessments

Several measures of body composition were significantly related (*p* < 0.05) to 2018 CFO performance (Table 3). Of these, body fat percentage was the best predictor of 18.1 (R^2^ = 0.80, *p* < 0.001), 18.2a (R^2^ = 0.55, *p* = 0.001), and 18.3 (R^2^ = 0.62, *p* = 0.001). Lean arm mass best predicted 18.2b (R^2^ = 0.72, *p* < 0.001) and body density best predicted 18.4 (R^2^ = 0.59, *p* = 0.001) and 18.5 (R^2^ = 0.67, *p* < 0.001).

#### 3.3.2. Muscle Morphology

While several measures of muscle morphology were related to 18.1 and 18.2b performance, only vastus lateralis muscle thickness was related to the remaining workouts (Table 4). Vastus lateralis muscle thickness was negatively related to 18.2a performance and positively related to performance on 18.3–18.5. According to stepwise regression, measures of vastus lateralis muscle size were the best predictors for all workouts, except 18.3 and 18.5, which could not be significantly predicted by muscle morphology. Vastus lateralis cross-sectional area best predicted performance on 18.1 (R^2^ = 0.28, *p* = 0.044) and 18.2b (R^2^ = 0.89, *p* < 0.001), while vastus lateralis muscle thickness best predicted 18.2a (R^2^ = 0.27, *p* = 0.047) and 18.4 (R^2^ = 0.29, *p* = 0.039) performance.

### 3.4. Cardiorespiratory Fitness Assessments

Several measures produced from the graded exercise test and 3-min maximal cycling sprint were significantly related (*p* < 0.05) to 2018 CFO performance (Table 5). Of these, RCT was the best predictor of performance on 18.2a (R^2^ = 0.75, *p* = 0.001), 18.3 (R^2^ = 0.75, *p* = 0.001), 18.4 (R^2^ = 0.58, *p* = 0.006), and 18.5 (R^2^ = 0.70, *p* = 0.001). HR_Recovery_ was the best predictor of 18.1 performance (R^2^ = 0.54, *p* = 0.010) and anaerobic work capacity best predicted performance on 18.2b (R^2^ = 0.71, *p* = 0.001).

### 3.5. Isometric Mid-Thigh Pull Strength

Several measures of force production obtained from the isometric mid-thigh pull test were related to performance on 18.1 and 18.2b (Table 5). However, only relative RFD_Peak_ was significantly related to performance on 18.3–18.5 (ρ = 0.54–0.61, *p* < 0.05), and no measure of strength was related to performance on 18.2a. Of the significantly correlated variables, relative RFD_Peak_ was the best predictor of 18.3 (R^2^ = 0.30, *p* = 0.036), 18.4 (R^2^ = 0.31, *p* = 0.031), and 18.5 (R^2^ = 0.37, *p* = 0.016) performance. RFD at 250 ms best predicted 18.1 performance (R^2^ = 0.40, *p* = 0.011) and RFD at 150 ms best predicted 18.2b performance (R^2^ = 0.76, *p* < 0.001).

### 3.6. Prediction of 2018 CFO Performance

Of the best workout predictors from each variable type, body fat percentage was the best predictor of performance on 18.1 (R^2^ = 0.89, *p* < 0.001), 18.2a (R^2^ = 0.55, *p* = 0.001), and 18.3 (R^2^ = 0.62, *p* = 0.001). Further, variance explained in 18.1 performance could be improved by 4% with the addition of RFD at 250 ms (*p* < 0.001). Vastus lateralis cross-sectional area was the best predictor for 18.2b performance (R^2^ = 0.78, *p* = 0.009), and body density was the best predictor for performance in 18.4 (R^2^ = 0.77, *p* = 0.021) and 18.5 (R^2^ = 0.67, *p* < 0.001); however, variance explained in 18.4 could be improved by 19%, with the addition of total experience in regional competition (*p* = 0.009). These relationships are illustrated in Figure 1.

## 4. Discussion

The purpose of this investigation was to determine the influence of experience and physiological fitness on performance in the workouts of the 2018 CrossFit^®^ Open competition. Based on past evidence [6,7,8,9,10,11,12,13,14] and CrossFit^®^ training’s general goal of eliciting general physical preparedness [1,2], we hypothesized that relationships would exist between CFO performance and each type of independent variable collected in this study. The data supported this hypothesis as several measures from each category were found to be related to performance on each workout. However, the secondary hypothesis, that the influence of each physiological measure on performance would vary between workouts and be modified by experience, was only partially supported. While workout performance could be explained by a variety of predictors, some measure of body composition or muscle morphology was found to be the most consistent overall predictor for each workout examined, and experience only improved predictive ability in one workout (18.4). This study appears to be the first to utilize a comprehensive assessment of athlete competition and training experience, along with a wide array of physiological fitness measures, to predict performance in an actual CrossFit^®^ competition.

The primary finding of this study was that body composition was the strongest predictor of performance in nearly all workouts examined compared with any other measure. Except for 18.2b, where vastus lateralis cross-sectional area was the best overall predictor, each workout was best predicted by either body fat percentage or body density. It is difficult to state whether this observation is in direct contrast with previous investigations into the predictors of CrossFit^®^ performance [6,8,9,11,12,13,14], because those studies either did not consider body composition as a predictor [6,9,11,12,13,14] or limited the analysis to body mass and stature [8]. Regardless, simple observation of elite CrossFit^®^ athletes (i.e., those who compete in the Games) suggests that being lean is a prerequisite for success in the sport. Although potentially the consequence of 2018 CFO workout design, our data support this observation, as those athletes who possessed a lower body fat percentage or greater body density performed better in competition workouts. In general, each workout contained a gymnastic component (e.g., burpees, handstand push-ups or walks, muscle-ups, pull-ups, toes-to-bar) and prescribed relatively lighter resistance exercise loads (i.e., less than the athlete’s body mass) for multiple repetitions; the one-repetition maximum clean on 18.2b and the heavier deadlifts on 18.4 were the only exceptions to this design. A leaner athlete would possess less non-functional mass and expend less energy performing repeated movements (e.g., pull-ups) than an athlete who possess a greater percentage of fat mass [33]. Meanwhile, assuming proper hydration and ventilation, the leaner athlete would theoretically possess a greater ability to thermoregulate during exercise [33]. In either case, both suppositions suggest that the leaner athlete may be better equipped to sustain effort, which would have been important for these workouts, because their durations ranged from 7 to 20 min. That said, athletes should take caution when interpreting this finding. Solely focusing on lowering body fat percentage to improve sports performance is not recommended, because it may lead to several health complications, particularly in females, and reduced energy availability [34] often required for peak performance.

Except for 18.2b, all workouts were scored by repetitions completed within the time limit or time-to-completion. While the athletes may have employed several unique strategies to earn the best score, ultimately, any strategy would have attempted to optimize workout density (i.e., completing work in the shortest amount of time). Within this context, an athlete who can sustain effort at a higher intensity and for a longer duration would earn a better score. Consistent with this hypothesis, VO_2_ at RCT was found to be the most consistent predictor among the cardiorespiratory fitness measures. RCT is thought to demarcate the point in which exercise transitions from ‘heavy’ to ‘severe’ [35,36]. Theoretically, athletes who reach this point later (i.e., at higher values of oxygen consumption) should be able to sustain a higher work rate. While it is surprising that neither RCT or any other cardiorespiratory variable were found to be the best (or second-best) overall predictor of performance on any workout, these findings agree with the literature. Outside of one study that only examined indices of aerobic and anaerobic fitness [11], these measures, though related to performance, have rarely been found to be the best predictors of CrossFit^®^ performance when other factors were also considered [8,9,12,14]. In previous studies, aerobic capacity was the best predictor in only one of four workouts [12], the second-best predictor (to experience) in one of two workouts [9], the third most influential variable for overall performance across five workouts [14], or not influential on performance across four workouts [8]. Although the reasons for this are unclear, these findings could be related to training specificity. With one exception [12], CrossFit^®^ training modalities have never been matched to the modalities used to assess cardiorespiratory fitness. Dexheimer and colleagues (2019) measured aerobic capacity on a treadmill, and then found it to be the best predictor of the benchmark workout “Nancy”, which involves five rounds of 400-m sprinting and 15 overhead squats. Another possible explanation may be related to the continuous nature of a typical aerobic capacity test, an “all-out” sprint test, or even a Wingate Anaerobic Threshold test. Although continuous effort seems to be advantageous to CrossFit^®^ performance, it still involves necessary breaks, as the athlete transitions between exercises or between eccentric and concentric phases of a lift. It is known that transitioning between intensities and modalities affects energy expenditure [37], and, thus, any test that requires continuous effort may not adequately reflect ability in this sport.

Vastus lateralis cross-sectional area being more predictive of 18.2b performance than any strength measure is interesting. The present study utilized an isometric mid-thigh pull test and self-reported performances in traditional power and Olympic lifts to quantify strength and predict CFO performance. Although strong relationships were observed, logically, these measures should have been the most influential on 18.2b performance, but they were not. Instead, a limited (to the middle region of one muscle involved in knee extension) measure of muscle size was the best predictor of 1-RM clean. However, there are a few explanations that may have individually, or by some combination, influenced this outcome. For instance, the self-reported measures of strength and isometric mid-thigh pull performance could be viewed as overestimates of 18.2b capability, due to their preceding events being markedly different. During the CFO, athletes were required to complete 55 repetitions in the dumbbell squat and bar-facing burpee exercises, before finding their 1-RM clean [20]. This was not the case for the mid-thigh pull test and most likely not the case for when the self-reported measures of strength were accomplished. Isometric mid-thigh pull performance may also be viewed as an overestimate, because force produced during the assessment exists on a different point of the force-velocity curve [38]; a point where greater force is thought possible compared with any lift involving concentric motion. Indeed, the relationships between peak force and RFD values obtained from this test and Olympic lifting performance have varied among weightlifters [39] and are unclear in other athletes [40,41]. Yet another explanation may be related to the meaning of the cross-sectional area value obtained for skeletal muscle. Though muscle size has often been positively associated with strength [42,43,44,45], its measurement via ultrasound may be indicative of a variety of muscular adaptations that are not necessarily indicative of force production [44]. In a recent review, Haun and colleagues (2019) defined the term ‘sarcoplasmic hypertrophy’ to represent changes in muscle volume that are accompanied by adaptations to intramuscular structures that are primarily involved with its metabolic functions. Such adaptations, in addition to strength, would have likely been useful for the pre-fatigued athletes when attempting a 1-RM clean during 18.2b.

Describing CrossFit^®^ experience has been an evolving process for predictive studies. Initially, Bellar and colleagues (2015) recruited individuals based on whether they had more than one year of experience, or were simply naïve to CrossFit^®^ training. Although experience could have been further defined by the level of competitive experience (e.g., local versus regional/Games^TM^), ultimately, the one-year distinction was found to be the best predictor of two novel workouts. Around the same time, another study utilized the one-year experience criteria, which actually translated to 3.9 ± 2.0 years of experience, but they did not consider this factor when predicting performance in common benchmark workouts [8]. Likewise, a later study expanded on the one-year criteria by describing participant percentile rank in the common benchmark workouts, but, again, did not consider experience as a predictive factor [12]. Most recently, Schlegel et al. (2020) utilized competitive ranking to recruit participants and then predicted rank based on self-reported performance in a variety of lifts and bodyweight exercises. The authors further described the actual years of experience and training habits of the participants, though these did not appear to be used as predictive variables. Thus, the present study appears to have taken the largest step forward in describing CrossFit^®^ training experience, and considering its influence on workout performance. Our findings suggest that broad and limited definitions of experience may not be useful for predicting CFO performance. Years of CrossFit^®^ and resistance training experience were not related to performance in any workout, and training regularity was only associated with two workouts (18.2b and 18.5). Rather, athletes who possessed more competition experience at various levels (i.e., CFO, regionals, and the Games) and ranked higher in past CFO competitions produced better scores on each 2018 CFO workout. The reasons for why competition experience could explain between 31% and 63% of variance in CFO workout performance, and was more important than training experience, are not clear. It could be indicative of learned competition strategies, or simply a modifying variable; regional experience was found to be the second-best predictor of 18.4 performance. A key limitation to this study was the smaller sample size restricting our ability to examine only 1–2 predictor variables at a time, and making it less likely for certain variables to be sufficiently represented (e.g., experience as an individual competitor in the Games, knowledge of current ability in the workouts, and strength measures that were self-reported). Future investigations would be better equipped to examine the interrelationships of multiple predictors of CFO performance with a larger sample.

## 5. Conclusions

When it comes to prediction of CrossFit^®^ performance, there appears to be a very clear difference between performance measured in the laboratory and competitive settings. The laboratory setting is useful for controlling several extraneous confounds, but it does not sufficiently emulate the competitive environment. In CFO competition, athletes are free to complete workouts in their normal training facility and in the presence of other competitors. They benefit from both intrinsic and extrinsic motivation factors in a familiar setting. These two environments have been shown to elicit very different responses in the same physiological measures following CrossFit^®^ style workouts [15,46,47]. Thus, it is not surprising that the results of this study differ from previous studies that have aimed to predict CrossFit^®^ performance. Collectively, past works suggest that performance is dependent upon a variety of fitness parameters, and based on those reports, athletes might attempt to train for multiple, competing fitness outcomes. Although the present study does support the notion that multiple fitness parameters are related to performance, our data suggest that high-level competitive experience, body fat percentage, vastus lateralis cross-sectional area, VO_2_ at RCT, and RFD were the most common predictors of CFO performance. Of these, body fat percentage (or body density) was generally the most important. Although potentially limited to the workouts investigated in this study, these data suggest that athletes should focus on maintaining a healthy ratio of fat mass to lean mass to maximize performance and emphasize training intensity as a secondary focus. Higher resistance training loads are known to influence adaptations in strength and muscle size in trained adults [48], while more-intense continuous and intermittent activity may have a more direct impact on the ability to sustain effort at a faster pace [37,49]. Finally, athletes who are exposed to higher levels of competition appear to have an advantage in the CFO over those who do not.

## Figures and Tables

**Figure 1 sports-08-00102-f001:**
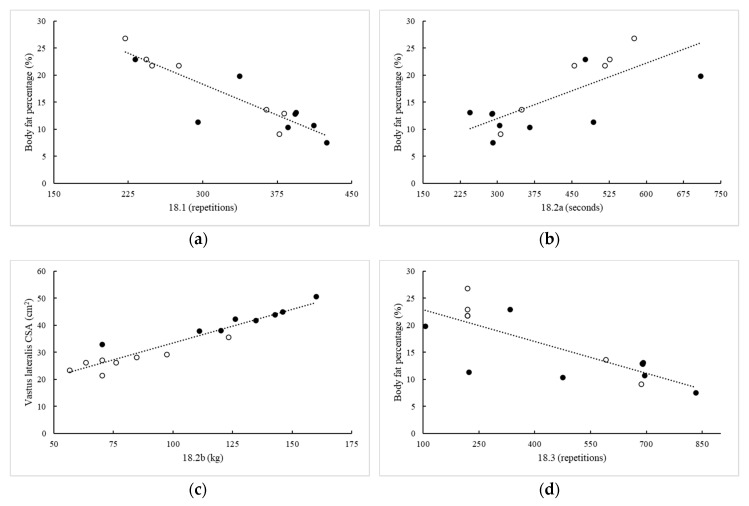
Best predictors of performance on CFO workout (**a**) 18.1, (**b**) 18.2a, (**c**) 18.2b, (**d**) 18.3, (**e**) 18.4, and (**f**) 18.5. Black, closed circles = men; Open circles = women.

**Table 1 sports-08-00102-t001:** Descriptions and athlete performance in 2018 CrossFit^®^ Open (CFO) workouts; mean ± SD (range).

Workout	Duration	Details	Score	Rank
18.1 (repetitions)	20-min AMRAP	8 toes-to-bar, 10 dumbbell hang clean and jerks (50-lbs; 35-lbs), and 14-calorie row	333 ± 70 (222–425)	25,690 ± 44,001 (262–174,849)
18.2a (seconds)	12-min time limit	1-2-3-4-5-6-7-8-9-10 repetitions of dumbbell squats (50-lbs; 35-lbs) and bar-facing burpees	412 ± 130 (245–709)	27,679 ± 39,542 (70–149,960)
18.2b (kg)		1 repetition-maximum clean	103 ± 34 (57–160)	22,972 ± 34,501 (3–139,015)
18.3 (repetitions)	14-min time limit	2 rounds of 100 double-unders, 20 overhead squats (115-lbs; 80-lbs), 100 double-unders, 12 ring muscle-ups, 100 double-unders, 20 dumbbell snatches (50-lbs; 35-lbs), 100 double-unders, and 12 bar muscle-ups	466 ± 239 (106–833)	25,297 ± 34,983 (74–126,714)
18.4 * (repetitions·s^−1^)	9-min time limit	21-15-9 repetitions of deadlifts (225-lbs; 155-lbs) and handstand pushups followed by 21-15-9 repetitions of deadlifts (315-lbs; 205-lbs) and 50-ft handstand walk	0.191 ± 0.083 (0.039–0.307)	27,278 ± 37,637 (293–133,969)
18.5 (repetitions)	7-min time limit	3-6-9-12-etc. repetitions of thrusters (100-lbs; 65-lbs) and chest-to-bar pull-ups	116 ± 33 (52–166)	20,899 ± 33,260 (317–132,118)

* The final score for 18.4 was officially recognized as either time to completion or the number of repetitions completed within the 9-min time limit. Since one athlete in this study completed the workout, it was necessary to create a uniform metric for analysis. Thus, all 18.4 scores were converted into repetitions per second.

**Table 2 sports-08-00102-t002:** Relationships between experience, self-reported fitness, and 2018 CFO performance.

	Mean ± SD	18.1	18.2a	18.2b	18.3	18.4	18.5
Experience							
Resistance training (y)	12.4 ± 7.2	0.03	0.22	0.19	−0.25	−0.33	−0.21
Regular resistance training (y)	9.8 ± 6.9	0.44	−0.44	0.80 *	0.41	0.40	0.54 *
Regular CrossFit (y)	4.8 ± 4.3	0.43	−0.37	0.30	0.43	0.36	0.37
2017 Open Rank	15,205 ± 17,823	−0.62 *	0.39	−0.30	−0.50 *	−0.49	−0.51 *
Highest previous Open rank	11,994 ± 16,152	−0.77 *	0.53 *	−0.42	−0.62 *	−0.55 *	−0.57 *
CrossFit Open (y)	3.3 ± 1.7	0.53 *	−0.56 *	0.29	0.67 *	0.62 *	0.54 *
Individual Regions (y)	0.3 ± 0.6	0.50 *	−0.41	0.34	0.51 *	0.46	0.50 *
Team Regions (y)	0.8 ± 1.3	0.44	−0.41	0.35	0.53 *	0.68 *	0.51 *
Total Regions (y)	1.1 ± 1.6	0.54 *	−0.49	0.41	0.62 *	0.72 *	0.60 *
Individual Games (y)	0.2 ± 0.8	0.30	−0.22	0.25	0.26	0.20	0.25
Team Games (y)	0.6 ± 1.0	0.43	−0.40	0.25	0.54 *	0.70 *	0.51 *
Total Games (y)	0.8 ± 1.3	0.50 *	−0.44	0.33	0.56 *	0.65 *	0.54 *
Self-reported fitness							
Back squat (kg)	151 ± 41	−0.14	0.20	0.72 *	−0.37	−0.05	−0.08
Clean and Jerk (kg)	107 ± 21	0.03	−0.32	0.92 *	0.15	0.42	0.35
Snatch (kg)	86.4 ± 17.3	0.09	−0.42	0.97 *	0.27	0.47	0.40
Deadlift (kg)	185 ± 38	0.05	0.15	0.82 *	−0.03	−0.28	−0.33
Fran (min)	3.8 ± 1.9	0.02	0.76	−0.81	−0.54	−0.78	−0.77
Grace (min)	1.9 ± 0.7	−0.27	0.10	−0.77	−0.09	−0.62	−0.18
Helen (min)	9.0 ± 1.5	−0.85 *	0.67	−0.19	−0.57	−0.85 *	−0.76

* Significantly (*p* < 0.05) related to CFO performance.

**Table 3 sports-08-00102-t003:** Relationships between body composition and 2018 CFO performance.

	Mean ± SD	18.1	18.2a	18.2b	18.3	18.4	18.5
Height (cm)	171 ± 12	0.18	0.07	0.51 *	0.03	−0.05	−0.10
Weight (kg)	78.0 ± 16.2	0.40	−0.07	0.68 *	0.16	0.04	0.11
BMI (kg·m^−2^)	26.5 ± 3.4	0.49	−0.25	0.57 *	0.30	0.17	0.35
Total body water (L)	48.2 ± 11.5	0.54 *	−0.25	0.75 *	0.37	0.23	0.29
Body density (kg·L^−1^)	1.059 ± 0.013	0.86 *	−0.70 *	0.60 *	0.77 *	0.77 *	0.82 *
Bone mineral content (kg)							
Total	3.22 ± 0.62	0.53 *	−0.21	0.71 *	0.35	0.21	0.29
Arms	0.492 ± 0.132	0.58 *	−0.28	0.76 *	0.42	0.28	0.34
Legs	1.19 ± 0.28	0.52 *	−0.24	0.75 *	0.37	0.24	0.31
Trunk	1.005 ± 0.182	0.49	−0.24	0.73 *	0.33	0.23	0.34
Lean mass (kg)							
Arms	8.37 ± 2.70	0.61 *	−0.36	0.85 *	0.44	0.36	0.42
Legs	20.8 ± 4.9	0.58 *	−0.29	0.81 *	0.38	0.29	0.34
Trunk	29.3 ± 6.4	0.64 *	−0.34	0.81 *	0.45	0.35	0.41
4-compartment model							
Body fat percentage (%)	15.8 ± 6.2	−0.89 *	0.74 *	−0.66 *	−0.78 *	−0.72 *	−0.81 *
Fat-free mass (kg)	66.6 ± 15.8	0.61 *	−0.28	0.78 *	0.39	0.28	0.35
Fat mass (kg)	12.1 ± 4.6	−0.64 *	0.71 *	−0.26	−0.67 *	−0.70 *	−0.75 *

* Significantly (*p* < 0.05) related to CFO performance.

**Table 4 sports-08-00102-t004:** Relationships between muscle morphology and 2018 CFO performance.

	Mean ± SD	18.1	18.2a	18.2b	18.3	18.4	18.5
Muscle thickness (cm)							
Rectus femoris	2.73 ± 0.49	0.41	−0.37	0.78 *	0.32	0.32	0.32
Vastus medialis	3.78 ± 0.85	0.44	−0.36	0.82 *	0.33	0.34	0.34
Vastus lateralis	1.96 ± 0.48	0.57 *	−0.58 *	0.76 *	0.54 *	0.56 *	0.54 *
Biceps brachii	3.49 ± 0.85	0.50	−0.26	0.70 *	0.29	0.27	0.32
Triceps brachii	2.73 ± 0.61	0.27	−0.24	0.47	0.18	0.25	0.29
Cross-sectional area (cm^2^)							
Rectus femoris	13.2 ± 4.3	0.46	−0.26	0.78 *	0.30	0.28	0.31
Vastus medialis	24.7 ± 6.9	0.53 *	−0.37	0.81 *	0.37	0.45	0.40
Vastus lateralis	34.3 ± 8.8	0.56 *	−0.45	0.95 *	0.48	0.43	0.49
Biceps brachii	12.1 ± 6.3	0.36	−0.23	0.67 *	0.17	0.16	0.24
Triceps brachii	13.1 ± 5.8	0.53 *	−0.28	0.71 *	0.30	0.25	0.34
Corrected echo intensity (au)							
Rectus femoris	125 ± 36	−0.44	0.18	−0.50	−0.32	−0.19	−0.23
Vastus medialis	117 ± 29	−0.49	0.21	−0.39	−0.38	−0.23	−0.23
Vastus lateralis	125 ± 36	−0.43	0.22	−0.46	−0.39	−0.23	−0.25
Biceps brachii	137 ± 34	−0.23	−0.01	−0.15	−0.13	0.02	0.01
Triceps brachii	99 ± 36	−0.48	0.25	−0.44	−0.40	−0.23	−0.29

* Significantly (*p* < 0.05) related to CFO performance.

**Table 5 sports-08-00102-t005:** Relationships between physiological fitness measures and 2018 CFO performance.

	Mean ± SD	18.1	18.2a	18.2b	18.3	18.4	18.5
Graded exercise test							
HR_Peak_ (bpm)	174 ± 13	−0.11	−0.06	0.02	0.12	0.10	0.11
HR_Recovery_ (bpm)	159 ± 15	−0.36	0.20	−0.23	−0.26	−0.25	−0.21
HR_Recovery_ (% of HR_Peak_)	−10.2 ± 4.9	−0.67 *	0.53	−0.66 *	−0.61 *	−0.60 *	−0.58 *
VO_2Peak_ (mL·kg^−1^·min^−1^)	48.0 ± 7.2	0.64 *	−0.69 *	0.54 *	0.70 *	0.55 *	0.71 *
RCT (mL·kg^−1^·min^−1^)	34.7 ± 4.9	0.57 *	−0.69 *	0.37	0.65 *	0.41	0.65 *
RCT (% of VO_2Peak_)	72.6 ± 6.7	−0.14	0.02	−0.19	−0.12	−0.24	−0.11
GET (mL·kg^−1^·min^−1^)	26.4 ± 4.2	0.47	−0.56 *	0.25	0.52 *	0.34	0.48
GET (% of VO_2Peak_)	55.6 ± 9.0	−0.09	0.03	−0.12	−0.09	−0.12	−0.10
3-min maximal cycling sprint							
Anaerobic work capacity (kJ)	37.8 ± 17.1	0.48	−0.33	0.81 *	0.43	0.41	0.47
Peak power (W)	1269 ± 575	0.52	−0.36	0.77 *	0.42	0.38	0.42
Critical power (W)	251 ± 67	0.68 *	−0.51	0.66 *	0.62 *	0.45	0.50
Isometric mid-thigh pull strength							
Peak Force (N)	1746 ± 473	0.56 *	−0.36	0.80 *	0.38	0.27	0.39
Relative Peak Force (N·kg^−1^)	2.35 ± 0.34	0.43	−0.49	0.45	0.34	0.32	0.47
RFD_Peak_ (N·sec^−1^)	1036 ± 606	0.55 *	−0.44	0.85 *	0.50	0.47	0.54 *
Relative RFD_Peak_ (N·sec^−1^·kg^−1^)	1.34 ± 0.63	0.54 *	−0.50	0.78 *	0.54 *	0.56 *	0.61 *
RFD_AVG_ (N·sec^−1^)	1568 ± 1728	0.40	−0.34	0.58 *	0.36	0.34	0.48
RFD at 30 ms (N·sec^−1^)	5452 ± 6279	0.34	−0.21	0.63 *	0.20	0.24	0.26
RFD at 50 ms (N·sec^−1^)	5973 ± 5811	0.41	−0.23	0.70 *	0.26	0.27	0.31
RFD at 90 ms (N·sec^−1^)	5874 ± 4266	0.49	−0.25	0.76 *	0.34	0.31	0.38
RFD at 100 ms (N·sec^−1^)	5763 ± 3884	0.50	−0.26	0.78 *	0.36	0.32	0.39
RFD at 150 ms (N·sec^−1^)	5415 ± 2717	0.56 *	−0.41	0.87 *	0.45	0.43	0.47
RFD at 200 ms (N·sec^−1^)	5304 ± 2441	0.62 *	−0.43	0.82 *	0.46	0.43	0.45
RFD at 250 ms (N·sec^−1^)	4947 ± 2096	0.64 *	−0.38	0.80 *	0.43	0.38	0.44

* Significantly (*p* < 0.05) related to CFO performance.

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
