# Peer review of "Predictors of CrossFit Open Performance"

_sports, 2020, doi:10.3390/sports8070102_

Round 1

Reviewer 1 Report

This manuscript sort to determine the influence that experience, self-reported fitness level as well as measured physiological fitness has on performance in CFO competition.

At the beginning of the Introduction you explain the CrossFit Open competition and that each workout is scored. A short explanation of the scoring system would be helpful here.

The methods used and results are explained clearly.

Change "compared to" to "compared with" throughout the manuscript. Lines 419 & 484 for exmple. 

The Discussion describes the outcomes of the study and provides an interesting commentary on the precarious nature of predictors of performance, in particular reliability of self report.

This manuscript provides some interesting conclusions about the usefulness of predictions of performance in CrossFit. A clear difference between measures in the lab and in competition is interesting and provides scope for further investigation on non-physiological factors such as psychological state as stated by the authors.

Author Response

This manuscript sort to determine the influence that experience, self-reported fitness level as well as measured physiological fitness has on performance in CFO competition.

That is correct. Thank you for taking the time to review our work. We have addressed each of your concerns within the manuscript when applicable and have responded to each concern specifically in the space below.

At the beginning of the Introduction you explain the CrossFit Open competition and that each workout is scored. A short explanation of the scoring system would be helpful here.

Beginning on Line 36, we have added the following to provide a short explanation of the scoring system and advancement:

“Scores are commonly entered as repetitions completed, a completion (of prescribed work) time, weight lifted, or some combination of these. Regardless, athletes’ scores are then ranked on each week using a system where the best performance receives the lowest rank (i.e., 1st place). Athletes’ ranks are tallied over the course of the CFO, and those who rank high enough after five weeks (i.e., have the lowest sum of their ranks from each workout) will advance in the competition.”

The methods used and results are explained clearly.

Thank you.

Change "compared to" to "compared with" throughout the manuscript. Lines 419 & 484 for exmple.

We have made these changes in the locations (the only two in the manuscript) you have identified here.

The Discussion describes the outcomes of the study and provides an interesting commentary on the precarious nature of predictors of performance, in particular reliability of self report.

This manuscript provides some interesting conclusions about the usefulness of predictions of performance in CrossFit. A clear difference between measures in the lab and in competition is interesting and provides scope for further investigation on non-physiological factors such as psychological state as stated by the authors.

Your last two comments are excellent examples of our intentions with this study. We acknowledge that are data cannot provide definitive answers to our primary research questions. However, we wanted to begin an earnest discussion about which factors are most important to CrossFit athletes. It is our hope that future studies may use our findings, as well as those from previous works, to 1) find more efficient and precise ways to quantify athletic ability in this sport, and 2) ultimately, determine which training targets are generally important for any competition, regardless of specific workout designs.

Reviewer 2 Report

This article aims to find the most important predictors for the performance in CrossFit competitions. High effort was made to measure a wide range of variables being related to basic skills in CrossFit, these are competition experience, biochemical markers in the blood, resting physiology, body composition, strength and cardiorespiratory fitness. Although limited by the small sample size results are well discussed and worth publishing. But before publication some important comments have to be considered!

CrossFit is a composition of basic athletic training exercises combined to selected basic skills of gymnastics like the handstand walk. Out of that basic fitness training competitions evolved leading to the question of the most important prerequisite to win. Because of the definition of CrossFit itself the answer will always depend on the selected work outs of the day. Additionally, the aim to maximize number of repetitions in basic training exercises lead to optimization of movement execution lacking a regulation of validity (burpees minimizing jump height, pull ups with swinging components). Therefor I would expect to see physiological measurements losing their strong relevance for prediction of CrossFit competition performance and a rising importance of competition experience (this comment addresses lines 510 to 519).

The hypothesis is well introduced. The selection of predicting variables is wide spread but sometimes variables miss reasoning of choice (e.g. resting measures)! As a weakness of the paper I would rate that a bunch of similar and interdependent variables are taken all together into the regression models of the subgroups without preselecting to minimize the number of calculated correlations. Preselecting would mean to defend the most meaningful variables in terms of measurement reliability, direct modulation of fitness levels and possibly well defined training goals. At the end of paragraphs it would be helpful to exactly address the (selected) variables being used and their expected influence on performance. This would also support the understanding of the orientation of correlation coefficients (e.g the inverse orientated 18.2a – seconds).

Please rethink interpretation of results (figure 1): While vastus lateralis CSA is an understandable predictor of WOD 18.2b (clean maximum), body fat percentage and body density are both indirectly hinting on well trained muscles without giving information of training outlines (maximum strength, short range strength endurance, speed?).

My recommendation to revise the paper is: reduce variables in every category referring to the strong inter-correlation sorting out by ease of interpretation and measurement reliability or last step of calculation proceedings.

Comments in detail:

Line 78:
please add the study of Martinez-Gómez, R., Valenzuela, P., Alejo L. et al. (2020). Physiological Predictors of Competition Performance in CrossFit Athletes. In: International Journal of Environmental Research and Public Health, 17(2020)10: 3699
also add information regarding this study to the discussion in line 450 ff.

Line 103: please add the number of men and women in your study (this information is only provided indirectly in figure 1)

Line 139: starting with table 1 directly following the heading: I would recommend to change the heading to: competition performance and self-reported fitness

Line 153: had not been completed by all of the athletes or none?

Line 166: please better define expectation: higher performance level with a lower ratio?

Line 177: what was assumed to predict with a higher resting energy expenditure (more muscle mass – better performance?)

Line 181: very interesting parameter! For all three paragraphs up to this here: think of posting the information of more detailed description of measurement techniques in [18] to the end of the paragraph.

Line 216 to 219: what was the expected information of bone mineral content and CrossFit performance?

Line 291: please add the used method of stepwise regression

Line 292: did you check for multicollinearity?

Comment to the method section: I would recommend reduce number of variables in the subgroups leaving out double information (muscle thickness, CSA; or body fat percentage, fat mass; reduce number of RFD calculation at different time points to an early and a late stage (e.g. at 50ms and 200ms – look for reasoning)

Line 305: table 2 – variables from “individual regions (years…)” to “total games (years)” show a very low mean; as described in line 114 half of the group has no experience (0 years) … means that the correlation seems to be questionable. I would recommend to omit these correlations

Line 316: Helen time correlates (rho = -0.85) to CFO 2018 performance 18.4: an explanation of variance of 91% seems to overestimate that relationship. Please check data!

Line 346: vast.lat muscle thickness was negatively related… add “negatively”

Line 383: last row of table 5: typo – it should be RFD at 250 ms

Typos: line 28… several ; line 434: who possesses; line 658: predictors of athleticism

Author Response

This article aims to find the most important predictors for the performance in CrossFit competitions. High effort was made to measure a wide range of variables being related to basic skills in CrossFit, these are competition experience, biochemical markers in the blood, resting physiology, body composition, strength and cardiorespiratory fitness. Although limited by the small sample size results are well discussed and worth publishing. But before publication some important comments have to be considered!

Thank you for donating your valuable time to provide a thorough critique of our manuscript. Although we disagreed with your recommendation to reduce the number of variables included in our study, we found many of your other recommendations to be thought provoking and beneficial to the quality of our manuscript. Below you will find our explanations regarding our points of disagreement, as well as descriptions of the areas we have revised in response to your concerns. We believe that you find both our explanations and revisions have sufficiently addressed your concern and improved the manuscript’s worthiness of publication.

CrossFit is a composition of basic athletic training exercises combined to selected basic skills of gymnastics like the handstand walk. Out of that basic fitness training competitions evolved leading to the question of the most important prerequisite to win. Because of the definition of CrossFit itself the answer will always depend on the selected work outs of the day. Additionally, the aim to maximize number of repetitions in basic training exercises lead to optimization of movement execution lacking a regulation of validity (burpees minimizing jump height, pull ups with swinging components). Therefor I would expect to see physiological measurements losing their strong relevance for prediction of CrossFit competition performance and a rising importance of competition experience (this comment addresses lines 510 to 519).

We do suspect that competition experience is a very important factor for success in CrossFit, and potentially more valuable than certain physiological traits. However, we felt that it would be presumptuous to state anything more than its importance being unclear at this time. CrossFit research in general has only recently started to provide more detailed descriptions of participant training experience, and only one other study has even considered its influence on performance using the most basic of descriptions (i.e., having more than one year of experience or not). Clearly, more studies are needed to elaborate on this topic before the importance of competitive experience can be accurately estimated.

The hypothesis is well introduced. The selection of predicting variables is wide spread but sometimes variables miss reasoning of choice (e.g. resting measures)! As a weakness of the paper I would rate that a bunch of similar and interdependent variables are taken all together into the regression models of the subgroups without preselecting to minimize the number of calculated correlations. Preselecting would mean to defend the most meaningful variables in terms of measurement reliability, direct modulation of fitness levels and possibly well defined training goals. At the end of paragraphs it would be helpful to exactly address the (selected) variables being used and their expected influence on performance. This would also support the understanding of the orientation of correlation coefficients (e.g the inverse orientated 18.2a – seconds).

We disagree that our inclusion of multiple, similar variables was a “weakness”. Rather, we believe it is a very important, and currently, a very much needed strength. When we set out to complete this project, there were less than a handful of studies that had previously sought to predict CrossFit performance. These studies were extremely focused in their selection of variables, and each identified a different aspect of fitness to be important. We believe that because these studies were not more inclusive in their inspection, the overall conclusion that can be made is that nearly all potential variables of interest are important to CrossFit. From a practical standpoint, this information is useless because it does not tell the coach or athlete which fitness components should be emphasized during training. Their only options are to systematically train for each component independently or attempt to train simultaneously for each, and neither option can be forecasted as being likely to succeed. The only way to begin to provide coaches and athletes with some direction is to take a more comprehensive approach. This was a point we made sure to emphasize in our Introduction.

The idea to take a comprehensive approach cannot be limited to variable “types”. It also extends to the characteristics that describe each variable type. For example, CrossFit lists “strength” as a fitness domain targeted by its training, and it defines strength as “the ability of the muscular unit, or a combination of muscular units, to apply force” (Glassman, 2011). This is a very general and incomplete definition, and we would guess that the most common interpretation is an individual’s “maximal force production” or “one-repetition maximum”. However, both interpretations only describe low-speed force expression, are limited to the movement (or exercise) performed, and do not describe force expressed at high velocities or anywhere else on the force-velocity continuum. This is an important distinction because success in sport, including CrossFit, appears to be dependent on force expression across the entire continuum (i.e., fine motor control movements, high-intensity efforts, high-velocity efforts, repeated sub-maximal efforts, etc.). Since previous studies could not provide any direction on which were more important, it became necessary to examine variables from multiple angles. The same argument could be applied to our measures of body composition, muscle morphology, and cardiorespiratory fitness. Though we do acknowledge that the Introduction fails to provide these details for each variable, doing so would have significantly and unnecessarily increased its length. Likewise, including in-depth descriptions in the Methods would have also increased each section’s length unnecessarily. That said, we have provided brief explanations for select variables (i.e., resting measures) where the rationale may not be obvious. We are confident that given the broad fitness aims of CrossFit, their inclusion as part of a more comprehensive approach to quantifying fitness will be apparent to readers. Further, many of these had been described in a previous publication (Mangine et al., 2020), and we address the importance of variables towards training in our Discussion.

Regarding pre-selection, though the assessments used in this study were quite common in the sports science literature, there was no evidence in the CrossFit literature available to guide us on such a procedure. Without pre-existing evidence, distinguishing the importance of say, RFD at 50ms and RFD at 250ms, would have been impossible before the study. These time epoch, and many between, have reportedly differential relationships to sports performance, and the wide variety of physical requirements though to be important in CrossFit only further clouds which should be expected to be most influential. Likewise, no study had included measures of body composition or muscle morphology as predictors of performance, and thus, we could not determine in advance, whether overall (i.e., body fat percentage) or more specific (e.g., lean mass, CSA, BMC, etc.) measures might be most relevant. Though each describe body composition in some capacity, they are all unique in both their quantification and as we have observed, their influence on performance. Thus, we elected to demonstrate the selection process through our statistical approach. As detailed in our manuscript, we began by reporting relationships between performance on each workout and variables within each classification. Statistically, this is a recognized and necessary step before regression analysis, where we selected significantly related variables to predict performance. By using the stepwise procedure, we funneled the large list of variables to the best predictor from each category. Finally, repeating the stepwise procedure where only the best predictors from each category were entered into the model, we were able to isolate the best overall predictor(s) for each workout. We did not attempt to develop an actual prediction equation because our sample was not sufficient to support more than 2 variables, and certainly not large enough to perform validation procedures. Nevertheless, future research will benefit from the foundation our data provides, and not necessarily have to include the same volume of variables.

Glassman, G. CrossFit training guide level 1. The CrossFit Journal: 2011.

Mangine, G.T.; Stratton, M.T.; Almeda, C.G.; Roberts, M.D.; Esmat, T.A.; VanDusseldorp, T.A.; Feito, Y. Physiological Differences Between Advanced Crossfit Athletes, Recreational Crossfit Participants, and Physically-Active Adults. PLoS One 2020, 14, e0223548, doi:10.1101/782359.

Please rethink interpretation of results (figure 1): While vastus lateralis CSA is an understandable predictor of WOD 18.2b (clean maximum), body fat percentage and body density are both indirectly hinting on well trained muscles without giving information of training outlines (maximum strength, short range strength endurance, speed?).

We are not clear on how you believe we should interpret our findings. While we do acknowledge that an athlete’s body fat percentage and body density provide some evidence of their training status, there are simply too many factors (i.e., genetics, diet and nutrition, training history) that limit the ability of these measures to be used to gauge the current training status of these athletes. Simply having a large amount of lean mass does not mean the individual is strong nor does having a low body fat percentage mean the individual is aerobically fit. Thus, we have taken a more conservative approach and discussed the most likely advantages of being lean in this sport.

My recommendation to revise the paper is: reduce variables in every category referring to the strong inter-correlation sorting out by ease of interpretation and measurement reliability or last step of calculation proceedings.

Your rationale for this suggestion is unclear. Each of the variables included in this study are quite common in the sports science literature and are relevant to athletes. Our data suggests that nearly all variables investigated in this study were related to performance on one or more of the workouts, and this is consistent with past findings. However, by using the stepwise procedure, we were able to statistically isolate the most influential variable(s) for each workout. This enabled us to conclude that while many variables were related, measures of body composition appeared to influence performance with the most consistency.

If we revised the manuscript to eliminate mention of a large percentage of the variables investigated, it would not change the final statistical outcome. However, doing so would affect the context of our conclusions, comparisons that could be made between our study and others, and it would reduce the amount of information we would be making available to readers. The fact is that relationships between many of these variables and CrossFit performance have not been previously investigated. Removing this information would affect the scientific record and reduce the value of this manuscript.

Comments in detail:

Line 78:

please add the study of Martinez-Gómez, R., Valenzuela, P., Alejo L. et al. (2020). Physiological Predictors of Competition Performance in CrossFit Athletes. In: International Journal of Environmental Research and Public Health, 17(2020)10: 3699

also add information regarding this study to the discussion in line 450 ff.

Thank you. We have added information related to this study to both our Introduction and Discussion.

Line 103: please add the number of men and women in your study (this information is only provided indirectly in figure 1)

We have made a brief revision to the first sentence of our participants section (2.1) to indicate that an equal number of men (n = 8) and women (n = 8) participated in this study.

Line 139: starting with table 1 directly following the heading: I would recommend to change the heading to: competition performance and self-reported fitness

We have revised the heading to section 2.3 to be “Competition performance and self-reported fitness”

Line 153: had not been completed by all of the athletes or none?

Occasionally, 1 or 2 participants would have recorded a value for one or more of the following parameters: Pull-ups, 400-m run, 5K run, “Filthy-50”, and “Fight-Gone-Bad”. Overall, however, most participants left these blank. We have revised this sentence to better reflect this observation.  

Line 166: please better define expectation: higher performance level with a lower ratio?

Resting concentrations of testosterone, IGF-1, and cortisol have all been reported to be affected by training. There may be a positive or negative effect on their respective concentrations that depends on the duration, intensity, volume, and overall stress of training. Collectively, their quantification might provide insight into protein metabolism, damage, fatigue, and recovery status, all of which may impact performance. Within the context of this acute study, and in line with the individualistic nature of hormone analysis, our expectation was that their concentrations or the TC ratio might act as a modifier to performance. That is, these variables would probably not be found to be the best indicators of performance, but they had the potential of being strong secondary predictors. For instance, an athlete with highly developed physical and physiological characteristics may enter the competition following a stressful training period (identified by high levels of cortisol and low levels of testosterone). Although they might still outperform less-skilled competitors, their performance might have been somewhat impaired due to accumulated fatigue, damage, etc. While we did see this in our data, our observation of the TC ratio being positively related to 18.2b performance represents the alternate possibility. That is, a higher ratio of testosterone to cortisol would imply a more anabolic (or better recovered) environment. This would seem to be an advantage for attempting a 1-RM clean immediately following 18.2a.

Though we would like to avoid overextending this manuscript with in-depth explanations of each variable of interest, we have added a line in section 2.4.1 to honor your request. We agree that our thought process of how training would have impacted hormone concentrations, which in turn could have affected performance, may not have been readily apparent to readers.

Line 177: what was assumed to predict with a higher resting energy expenditure (more muscle mass – better performance?)

In part, more muscle mass would require greater resting energy expenditure, and more muscle mass might be an advantage for performance. However, the reverse may also be true where too much muscle mass may be a disadvantage when it comes to repetitive efforts and gymnastic exercises. Resting energy expenditure, like resting hormone concentrations, may also be positively or negatively affected by training, as well as nutritional and dietary habits, all of which could in turn impact performance. Again, we did not expect this variable to be a primary predicting variable, but it had the potential of being a modifier. We have added a brief statement in section 2.4.2 to assist the reader in better understanding our rationale for including this variable.

Line 181: very interesting parameter! For all three paragraphs up to this here: think of posting the information of more detailed description of measurement techniques in [18] to the end of the paragraph.

We have done so for sections 2.4.1 and 2.4.2 but not for muscle morphology (2.5.1). We felt that doing so here would cause a greater disruption in the flow of the paragraph and that this paragraph was best left in its current format.

Line 216 to 219: what was the expected information of bone mineral content and CrossFit performance?

Skeletal mass and density are indicators for a greater ability to produce force (Schipilow et al., 2013). Force produced by muscle is transferred through the skeleton. If skeletal mass/density is not sufficient to withstand the force produced by muscle, theoretically, force would be lost or worse, injury might occur. Although we directly examined muscle mass and force production in our study, there are limitations to these measures that we included in our discussion. The inclusion of this measure examines the role of strength from a different angle. Additionally, skeletal mass played a role in the calculation of body fat percentage. It was possible that body fat percentage was important to performance, but when examined through the light of its components, one such as bone density could be more important. This was not the case, but that was a secondary line of thinking.

Schipilow J, Macdonald H, Liphardt A, Kan M, Boyd S. Bone micro-architecture, estimated bone strength, and the muscle-bone interaction in elite athletes: an HR-pQCT study. Bone. 2013;56(2):281-9.

Line 291: please add the used method of stepwise regression

We are not sure what you mean here. We stated that stepwise regression was used in our statistical analysis section. In SPSS, this is only carried out in one way; there are no alternative methods when the “stepwise” method is selected.

Line 292: did you check for multicollinearity?

Multicollinearity would only be an issue if we were attempting to predict performance from more than one variable. Due to our sample size, we purposefully limited our observations to the top 2 best predictors, and there were only two instances (18.1 and 18.4) where more than one variable was included in the regression. For 18.1, the top two variables were body fat percentage and RFD at 250 ms, while for 18.4, they were body density and total regions experience. In both cases, the two variables were very far from being similar. Additionally, the stepwise procedure does a good job of protecting against multicollinearity, as it will remove variables that do not significantly add to the prediction.

Comment to the method section: I would recommend reduce number of variables in the subgroups leaving out double information (muscle thickness, CSA; or body fat percentage, fat mass; reduce number of RFD calculation at different time points to an early and a late stage (e.g. at 50ms and 200ms – look for reasoning)

Your rationale for wanting us to remove information about significantly related variables is not clear. Though both are metrics of muscle size, muscle thickness and CSA do not provide duplicate information. On several occasions, relationships to performance in other sports have been reported from one of these measures and not the other. Their quantifications require different ultrasound modes, durations, and technician expertise. More importantly, neither have been previously investigated in relation to CrossFit performance. Likewise, body fat percentage and fat mass are also not the same, being relative and absolute values, respectively. These too have not been examined in relation to CrossFit performance. Finally, without having any previous evidence to direct our collection of different RFD time frames, limiting the examination to 50ms and 200ms would be questionable (i.e., there is not precedent for limiting this examination). Instead, reporting the variety of time epochs examined in this study is common, and limiting the examination to the two suggested would alter the findings. RFD at 150ms and 250ms were found to be the best predictors for 18.1 and 18.2b among strength measures. Removing these, as you suggest, would eliminate this information in favor of something that is less predictive.

Line 305: table 2 – variables from “individual regions (years…)” to “total games (years)” show a very low mean; as described in line 114 half of the group has no experience (0 years) … means that the correlation seems to be questionable. I would recommend to omit these correlations

A valuable aspect of this study that expands on the shortcomings of previous works is in regard to the concept of experience. In the past, studies have described experience in terms of training sessions per week and the duration of following such a regimen (usually for greater than 1 year). However, this does not inform the reader about experience quality. Reporting to the gym on three sessions per week for a year provides no indication of training volume, intensity, density, or variability. It provides no information about how those participants might compare physically or physiologically to similarly characterized individuals elsewhere. In contrast, ranking in the CFO, having qualified for (the now non-existent) regionals or the Games, and performance at these competitions provides a standard for all future comparisons. Removing these would negatively impact the scientific record.

In regards to the correlations, the fact that half of the participants possessed no experience at the regional level or beyond does not make the correlations questionable. Rather, the fact that strong correlations were still observed in this context speaks volumes about the importance of having regional or Games experience. Indeed, this experience was found to be more important than training experience according to regression analysis. And yet, another way to view this finding is that competition experience is the key aspect that requires further investigation. It is possible that regular participation in competitive events against high-level athletes is a better predictor of CFO performance than simply having had experience of progressing beyond this round. Additionally, we make no claim anywhere in this manuscript that our findings are definitive. Readers can easily ask the same questions you are asking about our findings. However, removing information from the manuscript may also prevent those questions from being asked. That would be a shame because the purpose of publication is to promote scientific communication.

Line 316: Helen time correlates (rho = -0.85) to CFO 2018 performance 18.4: an explanation of variance of 91% seems to overestimate that relationship. Please check data!

Although that does seem high, there are several similarities between the workouts. The duration of 18.4 involved a 9-minute time cap and according to their self-reports, the participants in this study averaged 9 minutes to complete “Helen”. There were also similarities in design and exercise components that likely made “Helen” a much better estimate of performance than the much shorter and simpler (in design) “Fran” and “Grace” workouts. It is also possible that because the data was self-reported, and knowledge may not have been currently applicable, the relationship was affected. Although a point about self-reported data had been made in our past submission, we have added a line (end of last paragraph in discussion) to further emphasize this point.

Line 346: vast.lat muscle thickness was negatively related… add “negatively”

Thank you. We have added “negatively” here.

Line 383: last row of table 5: typo – it should be RFD at 250 ms

Thank you. We made this correction.

Typos: line 28… several ; line 434: who possesses; line 658: predictors of athleticism

Removed “The” before “Several”